# Community-Acquired, Bacteraemic Acinetobacter Baumannii Pneumonia: A Retrospective Review of Cases in Tropical Queensland, Australia

**DOI:** 10.3390/tropicalmed8080419

**Published:** 2023-08-18

**Authors:** Timothy Riddles, Daniel Judge

**Affiliations:** Cairns Hospital, Cairns, QLD 4870, Australia; daniel.judge@health.qld.gov.au

**Keywords:** Acinetobacter, *Acinetobacter baumannii* complex, Bacterial pneumonia, community-acquired infections, bacteraemia, severe pneumonia, tropical infection

## Abstract

Background: Community-acquired *Acinetobacter* pneumonia (CAAP) typically presents with rapid progression to fulminant disease and is complicated by high mortality. Australian epidemiological studies are few. Methods: We conducted a retrospective study on bacteraemic cases of CAAP over twenty years (2000–2019) in North Queensland. Cases were selected on microbiologic, clinical, and radiographic parameters. Data on patient demographics were obtained, along with microbial, antibiotic, mortality and climatic data. Results: 28 cases of CAAP were included. Nineteen (67.9%) were male, twenty-three (82.1%) were Indigenous Australians, and the mean age was 45.9 years. Most presentations were of moderate to severe pneumonia (25/28 (89.3%)). Furthermore, 90% of cases had two or more risk factors. The strongest risk factors for CAAP were alcohol excess and tobacco use. No statistically significant difference in presenting severity, ICU admission or mortality was seen between dry- and wet-season disease. Dry-season disease accounted for 35.7% of cases. Overall mortality was 28.6%. Early use of meropenem or gentamicin reduced mortality irrespective of presenting severity (mortality 17.6%) Non-targeted antibiotic therapy was associated with a non-significant difference in mortality of 44.4%. Conclusions: Early administration of targeted antibiotics can mitigate a high mortality rate. The choice of antibiotic therapy for community-acquired pneumonia should be based on severity, risk factors and clinical suspicion of CAAP rather than seasonality.

## 1. Introduction

*Acinetobacter baumannii* is a ubiquitous environmental bacterium [1,2,3,4]. Unlike nosocomial strains, community-acquired strains of *A. baumannii* manifest with a more fulminant course [1,5,6]. Consequently, the mortality rate can exceed 60% [1,2,7,8,9]. Community-acquired strains exhibit less anti-microbial resistance compared to nosocomial strains [10,11]. The Australian *Therapeutic Guidelines* represent the main reference for evidence-based antibiotic prescribing in Australia. The most recent revision of the *Therapeutic Guidelines* recommends the use of carbapenems, or single-dose gentamicin, in moderate-to-severe pneumonia, where CAAP is suspected [12]. Suspicion of CAAP is reliant on presentation in the tropical wet season in patients with typical risk factors [10,12,13,14]. Adherence to an earlier version of these antibiotic guidelines has been shown to significantly reduce the mortality rate to as low as 11% [7].

Epidemiological studies of community-acquired *Acinetobacter* pneumonia (CAAP) in Australia are few [1,7]. Herein we report on a retrospective cohort study conducted in Northern Queensland, the first outside Australia’s Northern Territory.

## 2. Methods

### 2.1. Aims

To evaluate risk factors associated with community-acquired *Acinetobacter* Pneumonia in a North Queensland cohort.To assess the efficacy of antibiotic therapy in community-acquired *Acinetobacter* pneumonia.To evaluate how antibiotic prescribing practices in CAAP influence mortality.To investigate mortality rate and factors associated with mortality.

### 2.2. Study Design

This study was conducted at Cairns Hospital, a main referral centre servicing the Far North Queensland region, encompassing 380,748 km^2^ and servicing a population of approximately 280,000 people, 9% of whom identify as being of Aboriginal or Torres Strait Islander heritage [15]. 

We present a retrospective cohort study on bacteraemic cases of CAAP presenting to Cairns Hospital between January 2000 and December 2019 inclusive.

### 2.3. Participant Recruitment and Sample Collection

Inclusion required clinical, radiologic and microbiologic evidence of *Acinetobacter* pneumonia. Isolation of the *Acinetobacter baumannii complex* from blood was required for diagnosis, as sputum cultures alone may be insufficient to diagnose CAAP [16]. Criteria for community-acquired infection required patients to be free of hospital presentation within the preceding 72 h. 

*Acinetobacter baumannii* bacteraemic pneumonia was diagnosed with the following criteria: Blood culture positive for the *Acinetobacter baumannii complex*.Clinical features consistent with pneumonia.Radiographic evidence of pneumonia on chest radiograph or computed tomography scan.

Mortality was defined as death attributable to *Acinetobacter* pneumonia within 30 days of hospital admission. Pneumonia severity was characterized using SMART-COP score [17], based on information at the time of admission. Data were collected for potential risk factors for CAAP as defined by the literature [1,2,18,19,20,21,22], which included the presence of renal failure, diabetes mellitus, chronic pulmonary disease, current tobacco use, hazardous alcohol intake, malignancy, immunosuppression, and heart failure. Wet-season disease was determined by a high average monthly rainfall between the months of October and April at the patient’s town of residence, available from the *Australian Bureau of Meteorology* [23]. Laboratory identification of *Acinetobacter baumannii complex* was undertaken using *Biomérieux Vitex MS* (MALDI-TOF). This method may be imperfect for the identification of all *Acinetobacter* species but is generally considered reliable for use in clinical practice. Antibiotic sensitivities were determined using *Vitek*. The methodologies used to determine antibiotic sensitivities for the *Acinetobacter baumannii complex* were undertaken with EUCAST for meropenem, ciprofloxacin, gentamicin, and sulfamethoxazole-trimethoprim. CLSI was used for the remaining antibiotic sensitivities.

### 2.4. Definitions

Unless otherwise stated, *A. baumannii* refers to the *Acinetobacter baumannii complex.* Hypoxaemia was defined by SMART-COP score parameters with either SpO_2_ of <94% or pO_2_ of <70 mmHg. Shock was considered if the systolic blood pressure was recorded as <90 mmHg at the time of presentation. Immunosuppression was based on either chronic glucocorticoid use (>10 mg/day of prednisolone or equivalent), maintenance immunosuppressive therapy, or a known immunodeficiency syndrome. Targeted antibiotic therapy refers to the first-line antibiotics given at the commencement of treatment. This does not refer to antibiotics that may have efficacy but are given as part of an ‘oral-tail’, i.e., ciprofloxacin or trimethoprim-sulfamethoxazole. First-line targeted antibiotic therapy included intravenous antibiotics with established efficacy for CAAP, either gentamicin or carbapenem.

### 2.5. Statistical Analysis

Statistical analysis was performed using IBM SPSS Statistics package version 27. The Chi-square test for independence or Fisher exact tests were used for the comparison of two categorical variables. The Mann–Whitney U-test was used for the comparison of categorical with continuous variables where continuous variables were non-parametric. Statistical significance was defined as a *p*-value of <0.05.

### 2.6. Ethics

This project was granted ethics approval by the Far North Queensland Human Research Ethics Committee, Cairns Hospital (HREC/16/QCH/106-1080QA).

## 3. Results

There were 135 identified encounters of *Acinetobacter spp*. bacteraemia, of which 37 met inclusion criteria for community-acquired *Acinetobacter baumannii* pneumonia. One case of *Acinetobacter haemolyticus* pneumonia was excluded and a further eight cases were unavailable. Twenty-eight cases were available for the final inclusion (Figure 1). The overall mortality was 28.6% (8/28). Greater mortality was seen in patients who did not receive targeted antibiotic therapy from day one (45.5%), compared with those who received gentamicin or carbapenem on day one (17.6%).

The mean age of patients presenting with CAAP was 43 (36.8–51.3) (Table 1). Patients were more likely to be male (67.9%), Indigenous Australians (82.1%), and present in the wet season (64.3%). 

Clinically, patients presented with sputum production (47.8%) and pleuritic chest pain (64.3%) and were observed to be pyrexical (57.1%) and hypoxaemic (71.4%). Moreover, 89.3% of patients had moderate or severe pneumonia (SMART-COP) and often required admission to ICU (57.1%).

Biochemical abnormalities included neutrophilia (57.1%) and elevated C-reactive protein (95.5%) (Table 2). The median time to blood culture positivity for *Acinetobacter baumannii* was 10.7 h. Radiographically, 9 (32.1%) patients had disease limited to the left hemithorax, 12 (42.9%) had disease limited to the right hemithorax, and 6 (21.4%) had bilateral infiltrates. In total, 16 (57.1%) had multi-lobar disease. Pleural effusions were uncommon (7/28 (25%)). No pleural aspirates were performed.

Current tobacco use and hazardous ethanol consumption were the risk factors most frequently associated with CAAP (Table 1). Tobacco smoking was not associated with increased mortality (Table 3). Furthermore, 89.3% of individuals had two or more recognized risk factors for disease. Disease severity was not associated with the presence of multiple risk factors, nor the length of hospital stay. Non-targeted antibiotics for the treatment of CAAP on day one were found to confer the greatest risk for mortality amongst CAAP patients (50%) (Table 3). While a greater proportion of patients with mild-to-moderate CAAP on presentation died (4/10) than those with severe CAAP (4/18), there was no statistically significant difference in mortality between the two groups (χ^2^ = 0.742, df = 1, *p* = 0.320).

There was no statistically significant difference between the season of presentation (wet vs. dry) and the effect on presenting severity (Table 1) (F = 0.395, df = 26, *p* = 0.535) or mortality (χ^2^ = 0.16, df = 1, *p* = 0.901). Although there were more presentations to the ICU during the wet season, 10/16 (62.5%), this did not reach statistical significance (χ^2^ = 0.052, df = 1, *p* = 0.820). Of the three patients who died from fulminant sepsis within 24 h, one occurred in the dry season and two occurred in the wet season. An equal number of patients received gentamicin on day one in both wet and dry seasons, respectively, and five cases received meropenem on day one in both wet and dry seasons.

A greater proportion of patients received CAAP-targeted antibiotic therapy on day one if they had severe SMART-COP score (z = 1.854, df = 26, *p* = 0.064). There was no significant difference between day 1 antibiotic choice and the requirement for ICU (χ^2^ = 3.194, df = 26, *p* = 0.535). Additionally, there was a non-statistically significant difference in day 1 targeted antibiotic choice and mortality (χ^2^ = 2.530, df = 1, *p* = 0.200). Moreover, 45.5% (5/11) of patients who did not receive targeted antibiotics died, whereas only 17.6% (3/17) receiving targeted antibiotics on day 1 died. Of the total deaths, 50% (4/8) did not receive gentamicin or meropenem on day 1 (Table 4). All isolates demonstrated in vitro sensitivity to first-line intravenous antibiotics, gentamicin, and meropenem (Appendix A).

## 4. Discussion

The demographics of our cohort mirror previous Australian studies with an over-representation of Indigenous Australians, males, and middle-aged people [1,7]. The laboratory investigations represented in Table 2 are consistent with infection and are not discriminatory alone for the diagnosis of CAAP. Most patients had a multi-lobar infection and no tendency to a single hemithorax.

Hazardous alcohol intake and current smoking status were the most common risk factors. Although supported by other Australian studies [1,7], this contrasts with studies from Southeast Asia where excess alcohol consumption is less common [6,8,18,24]. The high proportion of patients who smoke in the cohort suggests tobacco smoking to be a risk factor for disease acquisition. Interestingly, a greater proportion of patients who died were non-smokers (4/5), as opposed to tobacco smokers (4/23). This discrepancy may be explained by confounder bias. The true factors to account for this are difficult to identify given the nature of the data and due to the retrospective study design. Possible explanations may be due to smokers being considered a higher-risk patient group, and thus being treated more aggressively. Smokers possibly present earlier in the disease course and thus receive more timely antibiotics. It is also possible that whilst it may be a risk factor for disease acquisition, it may not confer any additional risk for mortality in those with pneumonia. Prospective trials or the use of a control group may help minimise the effect of confounder bias to identify if this is a true effect. Cases of diabetes mellitus (DM) and chronic renal failure (CRF) were underrepresented in distinction to other studies [1,6,8,10,19,20]. As few patients had pre-morbid renal function or assessment for DM, it is possible these comorbidities are under-reported. Our cohort had similar presenting severity to a large Australian cohort from Darwin with median SMART-COP scores of 6.0 and 5.5, respectively [7]. Despite similar severity, we had fewer ICU admissions (57% vs. 80%) [7].

An earlier pilot study during the wet season in tropical Australia demonstrated the throat carriage of *Acinetobacter* in 10% of residents consuming excessive alcohol [18]. Ethanol impairs host immunity by adversely affecting phagocytosis, intracellular killing, cytokine production, antigen presentation and B-cell function [2,25]. Additionally, ethanol may hinder surfactant production, reducing anti-bacterial activity [25]. Secondly, ethanol may promote the micro-aspiration of colonised pharyngeal bacteria leading to nascent pulmonary infection [16]. We propose that ethanol may contribute to upper airway colonisation with *Acinetobacter*, risk of aspiration, and impaired immune response to infection.

A recent study from the Northern Territory has shown that community-acquired strains of *A. baumannii* harbour fewer virulence factors than nosocomial strains [26]. Additionally, no correlation was seen between strains harbouring more virulence factors and disease severity, ICU admission or mortality [26]. Therefore, host factors such as alcohol and tobacco consumption are likely more important in promoting disease transmission and impairing host response to infection [2,26,27]. Alcohol intoxication in itself may influence bacterial gene expression [26] and promote the growth of *Acinetobacter* [2], adding to the complex interplay between host and bacterial factors affecting virulence.

It is important to recognise the role of risk factors in disease acquisition, host response and clinical outcomes, although they have less discriminatory value in distinguishing CAAP from other causes of severe pneumonia or Gram-negative sepsis. This should be taken into consideration given that Australian antibiotic guidelines lean on risk factor identification for the purpose of rationalising antibiotic therapy.

Community-acquired *Acinetobacter baumannii* pneumonia is considered a tropical or sub-tropical wet season disease [2,7,21,27,28,29,30]. We observed over one-third of patients presenting with CAAP in the dry season. This observation is supported by a case-control series in Japan, which found no difference in the season with respect to the number of CAAP presentations [22]. Other case series have reported a wet season ‘peak’ but found no significant difference between seasons [9] or acknowledged there were reasonable case numbers in the dry season [1,8]. Prior studies suggest the seasonal predominance is due to the promotion of bacterial growth in warm, humid environments [27,28,30], although this does not explain the mechanism leading to initial nasopharyngeal colonisation. We speculate that similar to *Burkholderia pseudomallei* [31,32,33] *Acinetobacter* is liberated from soil in aerosols during wet weather. We, therefore, propose that the initial nasopharyngeal colonisation may occur during the wet season, although subsequent infection may occur later with aspiration [2,16] or during a relatively immunocompromised state. This is a proposed mechanism suggested by the authors and one which would require further evaluation to confirm. A prospective trial that assesses the genetic sequence of naso-pharyngeal isolates in asymptomatic carriers to those with disease manifestation with CAAP and to environmental sources would be a step towards supporting this hypothesis.

We found no significant difference in disease severity at presentation as defined by SMART-COP scores between wet and dry season disease (*p* = 0.535). Furthermore, there was no significant difference in ICU presentations between wet and dry seasons (*p* = 0.82). This is an important consideration as a determinant for considering empiric treatment of CAAP if it occurs during the wet season [12,13,14]. In the authors’ experience, CAAP is not typically considered, nor is it deliberately treated empirically in the drier months. Surprisingly, there was an equal proportion of gentamicin and meropenem use on day one in both dry- and wet-season diseases. This potentially reflects coverage for severe CAP or Gram-negative sepsis, rather than empirical treatment for suspected CAAP.

We suggest that the relatively improved mortality rate of 28.6% in our cohort, compared to previous studies [1,2,9,18,19], is a reflection of a higher proportion of moderate-to-severe CAP treated with targeted antibiotics for CAAP. If subjects were given gentamicin or meropenem on day one, they were less likely to die (17.6%) compared to those who did not receive targeted antibiotics on day 1 (44.4%). This is despite lower acuity illness, as measured by the median SMART-COP score in those who did not receive targeted antibiotic therapy (4 vs. 7). Although this rate did not reach statistical significance, the trend may be clinically significant. Furthermore, presenting severity of CAAP did not influence mortality (*p* = 0.320). Our mortality rate of 28.6% exceeded that of the Darwin cohort at 11% [7]. However, all patients in the Darwin cohort received timely antibiotics targeted to CAAP [7], compared to 61% of patients in our cohort. Evaluating patients who received targeted antibiotics on Day 1, only three died (17.6%) with a much-improved mortality rate. Furthermore, two of these three patients died within hours of arrival. This more likely reflects decompensated sepsis than the failure of antibiotic therapy per se. This highlights the importance of timely targeted antibiotic therapy in mitigating mortality associated with CAAP including in initially mild to moderate illness, and further supports the conclusions drawn from the Darwin study that mortality rates can be significantly improved with gentamicin or carbapenems [7].

We compared outcomes between the two frontline antibiotics, meropenem and gentamicin, to determine efficiency and patient outcomes. Unfortunately, we had insufficient data to make definitive statistical conclusions. Table 4 compares the patient groups who received either gentamicin or meropenem upfront or who received it later in the course of the disease, presumably either due to deterioration in disease severity or following the identification of *Acinetobacter baumannii* on culture. Mortality, hospital length of stay and disease severity on presentation are similar between both meropenem on day one and gentamicin on day one. The higher proportion of patients who received meropenem that required ICU (8/10 meropenem, 2/4 gentamicin) may reflect the prescribing practices amongst ICU clinicians or the difference in disease severity.

Furthermore, there was greater mortality with delayed access to meropenem compared with a delay to gentamicin (3/5 vs. 0/3). Limited data may diminish the strength of our conclusions, but importantly, a single dose of administered gentamicin may provide efficacy whilst awaiting confirmatory culture. This finding supports recent revisions of the *Therapeutic Guidelines,* which suggest an initial single dose of gentamicin as part of ‘triple therapy’ for moderate disease where CAAP is suspected [12].

### Limitations

Given the retrospective nature of this study, some data were limited, including data for SMART-COP scores where blood gas analysis was unavailable for some patients. This may have underestimated disease severity. It was presumed the presenting severity guided antibiotic decision making on day one. Although this does not appreciate the rapidity with which patients may deteriorate, this may explain some discordance with antibiotic choice, patient outcomes and the defined severity of illness. Strict definitions of hazardous ethanol intake could not be adhered to and were reliant on clinician discretion and documentation. We defined cases of community-acquired disease to be free of hospital presentation within the preceding 72 h. Given the retrospective nature of cases, there may be instances of community-onset disease that are not strictly community-acquired. However, antibiotic susceptibility profiling of cases suggests cases were community-acquired. As this study lacked a control group, it was not clear if the identified risk factors were specific for CAAP or were simply risk factors that predispose to infections or pneumonia in general. Cases excluded due to the inaccessibility of patient records were over-represented by peripheral hospitals, possibly contributing to selection bias. Furthermore, as this is a single-centre study, overall results may be less generalisable. However, we believe this still adds to the growing literature on CAAP given previous Australian studies have been limited to a single centre in the Northern Territory of Australia.

## 5. Conclusions

We report retrospectively on twenty-eight cases of community-acquired bacteraemic *Acinetobacter baumannii* pneumonia from North Queensland. The risk factors identified are similar to prior Australian studies, with middle age, male, and Indigenous Australians over-represented. The most commonly reported modifiable risk factors included hazardous alcohol intake and current smoking, with 89% having two or more risk factors. First-line antibiotics for *A.baumannii* are gentamicin or carbapenems. We propose that risk factors confer the greatest likelihood of sepsis arising from CAAP, and therefore irrespective of initial severity early use of first-line antibiotic therapy can reduce mortality. We advocate for the use of targeted antibiotics in suspected cases of CAAP where clinical suspicion should be based on risk factors, disease severity and clinical acumen, irrespective of the season at presentation. Unfortunately, these risk factors are not unique to CAAP and thus do not discriminate from other infectious aetiologies.

The authors agree with current antibiotic guidelines for wet-season CAP in tropical Australia. Although our study consists of relatively few numbers, we have demonstrated dry season CAAP in a third of overall cases. Supported by the data from our cohort, we would contend the use of single-dose gentamicin is considered in future guidelines for the empirical cover of dry season CAAP in suspected cases whilst awaiting culture confirmation. Single-dose gentamicin may also be considered in cases of mild pneumonia where the patient later deteriorates, particularly as the time to blood culture positivity in most instances is brief. In our experience, single-dose gentamicin is safe to prescribe even in renal impairment or sepsis. This change in clinic practice may help improve mortality in dry-season disease. Furthermore, it may help diminish the over-reliance on carbapenems and the development of subsequent antibiotic resistance this entails. Overall, mortality rates remain poor even in well-managed cases and thus ongoing vigilance by clinical caregivers in Australia is paramount.

## Figures and Tables

**Figure 1 tropicalmed-08-00419-f001:**
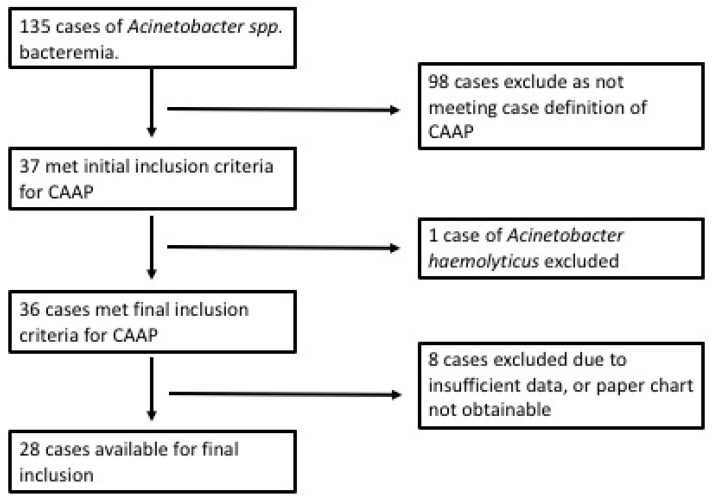
Case Selection.

**Table 1 tropicalmed-08-00419-t001:** Demographics, Presenting Features, and Presenting Severity of CAAP.

Patient Characteristic	Proportion of Patients, *n* = 28 (%)
Age in years, median, (IQR)	43, (36.8–51.3)
Male	19 (67.9)
Indigenous Australian	23 (82.1)
Resident of Cairns	9 (32.1)
Wet Season Disease	18 (64.3)
Dry Season Disease	10 (35.7)
**Clinical features**	
Pleurisy	18 (64.3)
Pyrexia (>38 degrees)	16 (57.1)
Muco-Purulent or Purulent Sputum	11/23 (47.8)
Blood-stained Sputum	11/23 (47.8)
**Severity**	
ICU admission	16 (57.1)
SMART-COP (moderate to severe)	25 (89.3)
SMART-COP, median and [IQR]	6, (4–8)
Hypoxaemia	20 (71.4)
Shock	10 (35.7)
Death	8 (28.6)
**Risk Factors for CAAP**	
Current tobacco Use	23 (82.1)
Hazardous Alcohol Consumption	20 (71.4)
Hazardous Alcohol & Tobacco Use	17 (60.7)
Chronic Lung disease	7 (25)
Heart Failure	4 (14.3)
Diabetes Mellitus	3 (10.7)
Chronic Kidney Disease	4 (14.3)
Immunosuppression	1 (3.6)
Malignancy	1 (3.6)
≥2 Risk Factors	25 (89.3)

IQR = Interquartile Range, ICU = Intensive Care Unit. N = 28, unless otherwise specified.

**Table 2 tropicalmed-08-00419-t002:** Biochemical Results on Presentation.

Test	Median [IQR]	Above Normal (%)	Below Normal (%)	Normal Reference Range
White Cell Count	11.8 (7.8–17.4)	14/28 (50)	5/28 (17.9)	4–11 × 10^9^/L
Haemoglobin	136 (121–146)	0 (0)	7/28 (25)	135–175/120–155 (M/F) g/L
Neutrophils	9.37 (5.46–15.5)	16/28 (57.1)	5/28 (17.9)	2–7.5 × 10^9^/L
Lymphocytes	0.53 (0.41–0.93)	0 (0)	21/28 (75)	1.5–4 × 10^9^/L
Platelets	177 (91.3–221)	1/28 (3.57)	12/28 (42.9)	150–400 × 10^9^/L
Albumin	29.5 (25–33.5)	0 (0)	21/28 (75)	35–50 g/L
C-Reactive Protein	154 (63–231)	21/22 (95.5)	0 (0)	<5 mg/L

IQR = Interquartile Range, M = Male, F = Female.

**Table 3 tropicalmed-08-00419-t003:** Risk Factors Related to Mortality.

Risk Factor (RF)	Number (%) of Those with RF Who Survived	Number (%) of Those with RF Who Died	*p*-Value
Shock	7/10 (70)	3/10 (30)	0.615
Chronic Lung Disease	4/7 (57.1)	3/7 (42.9)	0.411
Multi-Lobar Pneumonia	13/16 (81.2)	3/16 (18.8)	0.183
Non-targeted ABs D1	4/8 (50)	4/8 (50)	0.123
Hazardous Alcohol Consumption	14/20 (70)	6/20 (30)	0.589
Hazardous Alcohol & Tobacco Consumption	13/17 (76.5)	4/17 (23.5)	0.376
Tobacco Consumption	19/23 (82.6)	4/23 (17.4)	0.015 *
SMART-COP >/= 5 (Severe)	14/18 (77.8)	4/18 (22.2)	0.284
Multiple Risk Factors >2	18/25 (72)	7/25 (28)	0.652
Overall mortality	20/28 (71.4)	8/28 {28.6%}	

Non-targeted Antibiotics = antibiotics other than gentamicin or carbapenem, D1 = Day 1, defined as the date of admission to hospital, RF = Risk Factor, * Significantly more patients lived who were current smokers. Test statistic from Fisher exact test.

**Table 4 tropicalmed-08-00419-t004:** Risk Factors Related to Mortality.

Antibiotic Practice	No.	Died: Number (% Sample/% Total Deaths)	Died < 24 h	Mean LOS	ICU	Median SMART-COP on Admission
Meropenem D1	10	2 (7.1/25)	1	17.8	8	8
Gentamicin D1	4	1 (3.6/12.5)	1	16.5	2	6.5
Both D1	3	0	0	22.7	2	6
Total Mero/Gent D1	17	3 (10.7/37.5)	2	18.3	12	7
Delay to either	9	4 (14.3/50)	1	10	3	4
Delay Gentamicin	3	0	0	10.3	0	3
Delay Meropenem	5	3 (10.7/37.5)	0	10.6	3	4
Delay Both	1	1 (3.6/12.5)	0	6	0	2
Mero D1 delay gent	1	1 (3.6/12.5)	0	25	1	8
Gent D1 delay mero	3	0	0	21.7	2	6

LOS = Length of Stay, D1 = Day 1, Mero = Meropenem, Gent = Gentamicin, ICU = Intensive Care Unit.

## Data Availability

The data presented in this study are available upon request from the corresponding author. The data are not publicly available due to containing information that may compromise the privacy of participants.

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
