# Peer review of "Community-Acquired, Bacteraemic Acinetobacter Baumannii Pneumonia: A Retrospective Review of Cases in Tropical Queensland, Australia"

_tropicalmed, 2023, doi:10.3390/tropicalmed8080419_

Round 1
Reviewer 1 Report
Thank you for the submission. This paper is a paper describing community-acquired bacteremic Acinetobacter baumannii complex in Australia from 2000 - 2019, with the authors analyzing the data on 28 included patients. The manuscript is well-written, and here are some of my comments:
Major comments:
1. With the lack of a control group, it is difficult for the authors to comment further on what are the risk factors of CAAP identified in this study (because the baseline characteristics of your normal study population are not known). I understand it might be difficult to find a control group for this study design, therefore authors may consider including the lack of control as one of the limitations of your study.
2. The authors may want to consider revising Table 3. The presentation of Table 3 is confusing, as readers only know that there are differences in the percentages of patients with tobacco consumption in the survivor and non-survivor groups, but readers do not know whether it is a protective or risk factor. There should be four columns in this table with risk factors, the number(%) of those who survived with this risk factor, the number(%) of those who succumbed with this risk factor, and the p-value.
3. The authors actually found that 95.0% of patients who survived were smokers, while 50.0% of patients who died were smokers, with a p-value of 0.015. Are there any possible explanations to explain these findings?
4. One of the possible answers to major comment 3 is that there are confounders when the authors performed the analysis. Are there any significant differences between the survivors and non-survivors in terms of demographics, such as age, sex, race, symptoms, ICU admission, etc.? The authors may want to perform a multivariate analysis and see whether the findings noted in major comment 3 still hold true.
Minor comments:
1. Please include how your laboratory identifies Acinetobacter baumannii complex such as whether the identification is based on biochemical tests or MALDI-TOF, and which organisms your laboratory includes within Acinetobacter baumannii complex.
2. The cut-off of statistically significant p-value was not defined in your study (p <0.05?).
3. How many cases were excluded from your study? The numbers provided in your manuscript are incorrect. 36 cases, with 9 cases excluded, but 28 cases remaining for final inclusion.
4. Please recheck the percentage in Table 2. For Purulent sputum and blood-stained sputum, the percentages are incorrect.
5. Please provide the susceptibility profile summary of all the isolates within this study. Are they all susceptible to carbapenem and gentamicin? How does your laboratory perform antibiotics susceptibility testing (disc test, E-test, or microbroth dilution)?
Reviewer 2 Report
This study examined community-acquired Acinetobacter pneumonia (CAAP) in North Queensland, Australia, analyzing 28 cases over a twenty-year period. The findings revealed that early use of targeted antibiotics can reduce mortality, highlighting the significance of appropriate antibiotic therapy based on severity and risk factors, rather than seasonality. Despite the low number of included cases, the research is valuable for the community. However, there are some concerns:
To enhance the study, it would be beneficial for the authors to specify how they identified A. baumannii in the study. Biochemical tests or mass spectrometry alone may not provide high-confidence identification of A. baumannii among the Acinetobacter baumannii/calcoaceticus complex.
Regarding lines 200-204, while many Acinetobacter spp. are commonly found in soil, A. baumannii is typically present in soil or water samples due to previous human waste contamination. The parallel with B. pseudomallei may not be suitable for A. baumannii. This point should be discussed, and reference to Towner KJ's work (2009) can provide further insights. Lines 202-204 may be beyond the scope of this study, which focuses on community-acquired A. baumannii pneumonia.
Furthermore, lines 259-260 suggest that the strains are community-acquired based on antibiotic susceptibility, which contradicts the idea of soil contamination without prior human contamination. This inconsistency should be addressed.
It would be valuable if the authors could provide genetic information about the isolated strains to understand the heterogeneity among the cases.
Additionally, data on the familial or working environment of the patients would be beneficial. Exploring potential A. baumannii strains isolated from healthcare workers at Cairns Hospital could provide valuable clues for further investigation. According to the authors, the term "community-acquired disease" refers to cases that have not had any hospital presentation within the preceding 72 hours. However, there may be concerns regarding whether a 72-hours timeframe is sufficient to discount the possibility of recent hospital exposure.
Round 2
Reviewer 1 Report
Thank you for revising the previous manuscript on CAAP. The authors have addressed most of the issues previously raised. Here are several minor comments that the authors may want to address, then the manuscript should be ready for publication.
1. I agree with the authors' response that more people with tobacco consumption survived is likely due to confounder/ bias. Please include the explanation in the manuscript, and how it can be resolved in the future if further studies are performed in this area.
2. I agree with the authors' decision in putting the antibiotic sensitivity/ susceptibility profile Table in the supplementary material. The authors may want to include a line in your discussion, such as "All the strains of Acinetobacter baumannii complex included in this study were shown to be susceptible to meropenem and gentamicin, which are the first-line agents recommended for moderate-to-severe pneumonia."
3. A minor comment on Table 4, better use "Ticarcillin/Clavulanate" instead of Timentin.
Reviewer 2 Report
The section covering materials and methods should be expanded to provide readers with more comprehensive information. For instance, including additional details about the specific type of MALDI-TOF being employed would enhance clarity. Additionally, it's worth noting, as highlighted in both the literature and my earlier review, that MALDI-TOF might not be the optimal approach to confidently identify Acinetobacter baumannii. I refer you to the source: 10.1016/j.mimet.2015.09.006.
Moreover, the connection between A. baumannii's soil origin and its discussion in relation to existing literature appears somewhat speculative. A more robust analysis is needed in terms of A. baumannii reservoir identification, which is still a nascent field. Thus, I suggest that the introduction and conclusion be cautiously phrased in this regard.
